# Do We Really Need Message Passing in Brain Network Modeling?

Liang Yang [1]   Yuwei Liu [1]   Jiaming Zhuo [1]   Di Jin [2]   Chuan Wang [3]   Zhen Wang [4]   Xiaochun Cao [5]

## Abstract

Brain network analysis plays a critical role in brain disease prediction and diagnosis. Graph mining tools have made remarkable progress. Graph neural networks (GNNs) and Transformers, which rely on the message-passing scheme, recently dominated this field due to their powerful expressive ability on graph data. Unfortunately, by considering brain network construction using pairwise Pearson's coefficients between any pairs of ROIs, model analysis and experimental verification reveal that *the message-passing under both GNNs and Transformers can NOT be fully explored and exploited*. Surprisingly, this paper observes the significant performance and efficiency enhancements of the Hadamard product compared to the matrix product, which is the matrix form of message passing, in processing the brain network. Inspired by this finding, a novel Brain Quadratic Network (BQN) is proposed by incorporating quadratic networks, which possess better universal approximation properties. Moreover, theoretical analysis demonstrates that BQN implicitly performs community detection along with representation learning. Extensive evaluations verify the superiority of the proposed BQN compared to the message-passing-based brain network modeling. Source code is available at https://github.com/LYWJUN/BQN-demo.

[1]Hebei Province Key Laboratory of Big Data Calculation, School of Artificial Intelligence, Hebei University of Technology, Tianjin, China [2]Tianjin Key Laboratory of Cognitive Computing and Application, College of Intelligence and Computing, Tianjin University, Tianjin, China [3]School of Computer Science and Technology, Beijing JiaoTong University, Beijing, China [4]School of Artificial Intelligence, OPtics and ElectroNics (iOPEN), School of Cybersecurity, Northwestern Polytechnical University, Xi'an, China [5]School of Cyber Science and Technology, Shenzhen Campus of Sun Yat-sen University, Shenzhen, China. Correspondence to: Jiaming Zhuo <jiaming.zhuo@outlook.com>.

*Proceedings of the 42nd International Conference on Machine Learning*, Vancouver, Canada. PMLR 267, 2025. Copyright 2025 by the author(s).

## 1. Introductions

Brain network analysis provides a deep understanding of human brain organizations and assists in neurological disease diagnosis (Fornito et al., 2016; Cui et al., 2022). As a general-purpose language to model complex relationships, the research on graphs has a long history, ranging from graph theory (Bondy & Murty, 2008), to network science (Barabási, 2013), to network embedding (Cui et al., 2019). Along with the concept of the brain network, graph mining is widely used in brain network analysis, including concepts, characteristics, and models (Fornito et al., 2016; Chung, 2019). Network motif (Sporns & Kötter, 2004), small-world property (Bassett & Bullmore, 2006), and modularity (Meunier et al., 2009) are identified from the perspective of characteristics, while the persistent homology (Bendich et al., 2016), community detection (Garcia et al., 2018) and null model (Váša & Mišić, 2022) are employed to analyze brain network from the perspective of the model.

Graph Learning (Xia et al., 2021) plays an important role in diverse machine learning fields. (1) For tasks on graph data, Graph Neural Networks (GNNs), which combine graph topology and node attribute for representation, recently dominated this field (Wu et al., 2021b). They follow the message-passing scheme (Gilmer et al., 2017) by propagating node representation as a message over the graph. (2) For the domains without explicit graph structure, such as natural language processing (NLP) and computer vision (CV), implicit graph learning also shows amazing performance. The key component Multi-Heads Self-Attention (MHSA) in Transformer (Vaswani et al., 2017) and its variant ViT (Dosovitskiy et al., 2021) constructs a fully connected graph between all token pairs and performs a holistic aggregation (Section 3.3). In conclusion, the message-passing scheme dominates many machine learning fields.

Interactions between brain regions are regarded as the key factors for neural development and disorder analysis. Functional MRI (fMRI) provides valuable information for exploring connectivity by capturing correlations between signal sequences of brain regions. Thus, it is direct to analyze brain networks, which are often constructed using pairwise Pearson correlation coefficients between any pairs of ROIs, using powerful graph learning tools, such as GNNs (Li et al., 2021; Cui et al., 2022; Bessadok et al., 2023) and

Transformers (Kan et al., 2022b; Yu et al., 2024). Unfortunately, following model analysis and experiments motivate questioning this direct treatment in the brain network.

- GNN-based methods constructed node attributes from the brain network topology, which goes against GNNs' need for different types of information, and thus can't be fully exploited GNNs' characteristics (Fig. 1).
- Transformers employ the constructed brain network containing a holistic relationship between all ROIs, as initial token embedding. Thus the necessity to explore holistic relations with Transformers is weak (Fig. 1).
- Simple classifier with brain functional connectivity matrix as feature outperforms basic GNNs and Transformers (Fig. 2).

To enhance the performance of brain network analysis, the fundamental brain network operator should be updated. The recent breakthrough points out that the quadratic function can implement XOR logic operation and possesses better universal approximation properties compared to the linear function (Fan et al., 2018; 2020). Inspired by this, the quadratic network is employed in brain network analysis to obtain the Brain Quadratic Network (BQN). BQN iteratively performs Hadamard product/element-wise product between the representation in the previous layer and the initial representation to get a new representation. Theoretical analysis reveals that BQN is equivalent to the updating rule of a community detection objective function based on nonnegative matrix factorization (NMF) of the adjacency matrix. Since the NMF of the adjacency matrix is a widely-used community detection paradigm, the proposed BQN seeks representation, which can capture mesoscopic community structure to reflect brain functional modules. The main contributions of this paper are summarized as follows.

- We investigate the rationality of the widely adopted message-passing in the brain network analysis.
- We propose a simple and effective Brain Quadratic Network (BQN) as the fundamental operator, with superior computational efficiency.
- We provide rigorous theoretical analysis connecting the proposed BQN with community detection.
- The proposed BQN achieves new SOTA on widely-used brain datasets.

## 2. Related Work

Recent advancements in brain network analysis have led to a surge in the application of graph-based learning methods, especially Graph Neural Networks and Graph Transformers.

**Graph Neural Networks.** In recent studies, Graph Neural Networks (GNNs) have been leveraged to learn representations of brain regions, as well as the intricate patterns of functional connectivity within the brain. For example, BrainGNN (Li et al., 2021) utilizes ROI-aware GNNs and specialized pooling operators to identify critical nodes, improving the model's interpretability. BrainGB (Cui et al., 2022) systematically assesses various GNN designs on brain network data and introduces a practical pipeline that enhances GNN applications in brain research. FBNetGen (Kan et al., 2022a) constructs a learned task-oriented brain graph for downstream tasks. A-GCL (Zhang et al., 2023) proposes an adversarial self-supervised brain graph neural network by integrating graph contrastive learning with ROIs multiband information. However, these approaches often concentrate on local neighborhood aggregation, potentially neglecting the significance of interactions between non-adjacent brain regions.

**Graph Transformers.** Graph Transformers are utilized in the analysis of brain networks, with the objective of capturing the holistic interaction of brain regions. Brain-NETTF (Kan et al., 2022b) employs classical Transformer encoders to generate ROI embeddings based on Pearson correlation matrices, along with an orthogonal clustered readout. ALTER (Yu et al., 2024) has specifically designed a brain graph Transformer to capture long-range dependencies among brain regions. ContrastPool (Xu et al., 2024) emphasizes attention on ROIs and subjects to learn a contrast graph, guiding the generation of brain representations through graph pooling. BioBGT (Peng et al., 2025) captures the small-world architecture in brain graphs through a network entanglement-based technique, emphasizing the biological characteristics of brain structure.

However, many of these methods often conflate the concepts of brain functional connectivity matrices and region of interest (ROI) features. The correlation coefficient matrix serves a dual purpose: it acts as both the adjacency matrix of the brain graph and the feature matrix. This duality contradicts established principles in graph research, where topology and features are typically regarded as two distinct types of information. This observation prompts us to design a simple yet effective network architecture specifically tailored for brain functional connectivity.

## 3. Preliminaries

This section provides notations, problem definition, and the concepts of GNNs and Transformers.

### 3.1. Notations and Problem Definition

Brain network, which models the connectivity between ROIs, can be represented as a graph $G = (\mathcal{V}, \mathcal{E}, \mathbf{X})$, where $\mathcal{V} = \{v_1, v_2, ..., v_N\}$ stands for the collection of $N$ nodes, i.e., ROIs, $\mathcal{E}$ denotes the collection of edges between nodes, and $\mathbf{X} \in \mathbb{R}^{N \times D}$ represents the feature matrix with the

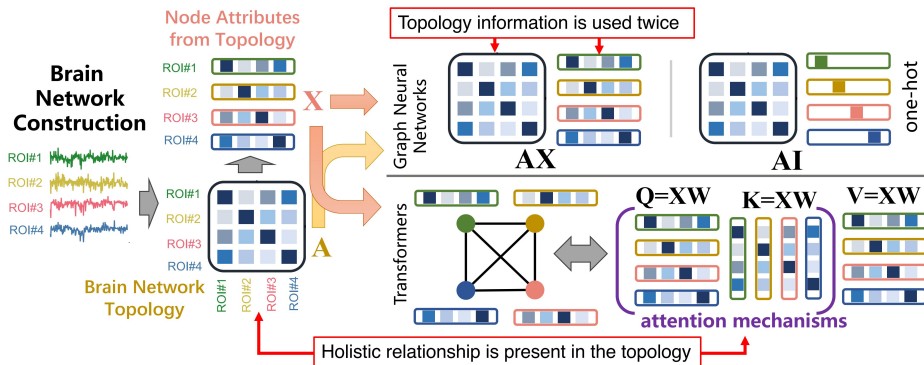

*Figure 1.* Brain network construction and the GNNs and Transformers on brain network analysis. In brain network construction, node attributes are from constructed brain network topology. Thus, GNNs utilize this information twice. The brain network already contains the correlation between all ROIs, while the attention mechanism also tends to learn this.

$i$-th row $\mathbf{x}_i \in \mathbb{R}^D$ as the feature of node/ROI $v_i$. The graph topology is represented as the adjacency matrix $\mathbf{A} = [a_{ij}] \in \mathbb{R}^{N \times N}$, where $a_{i,j}$ is the weight between nodes $v_i$ and $v_j$. $\mathcal{N}(v)$ denotes the neighbourhoods of node $v$. $\mathbf{D} \in \mathbb{R}^{N \times N}$ stands for the degree matrix with diagonal element $d_{vv} = \sum_{u \in \mathcal{N}(v)} a_{uv}$ as the degree of node $v$.

For brain network analysis tasks, a set of $L$ subjects' brain network $\mathcal{G} = \{G_1 \ldots G_L\}$ and the corresponding labels $\mathcal{Y} = \{y_1 \ldots y_L\}$, which indicates the presence of a disease, biological sex or other properties of the subject, are provided. Brain network modeling aims to learn from given data $\mathcal{G}$ and $\mathcal{Y}$ by designing a function $y = f(h(G))$, which is composed of a representation learning function $h_G = h(G)$ and a prediction function $y = f(h_G)$ based on the learned brain representation $h_G$.

### 3.2. Graph Neural Networks

Graph Neural Networks (GNNs) often follow the message-passing paradigm (Gilmer et al., 2017; Yang et al., 2022; Zhuo et al., 2023), by iteratively aggregating node representations from neighborhoods and combining them with the representation of itself. By denoting $\mathbf{H}^l$ as the collection of representations in the $l$-th layer, its $v$-th row $\mathbf{h}_v^l$, i.e., the representation of node $v$ in the $l$-th layer, can be obtained as

$$\hat{\mathbf{h}}_v^l := \text{Aggregation}^l(\{\mathbf{h}_u^{l-1} | u \in \mathcal{N}(v)\}), \quad (1)$$

$$\mathbf{h}_v^l := \text{Combination}^l(\mathbf{h}_v^{l-1}, \hat{\mathbf{h}}_v^l), \quad (2)$$

where $\text{Aggregation}(\cdot)$ and $\text{Combination}(,)$ denote the aggregation and combination modules, respectively. Following the above paradigm, classical GNNs, e.g., GCN (Kipf & Welling, 2017)-one of representative GNNs employ the weighted average function $\tilde{\mathbf{A}}\mathbf{H}^{l-1}$ to implement the above operations as

$$GCN: \quad \mathbf{H}^l = \sigma(\tilde{\mathbf{A}}\mathbf{H}^{l-1}\mathbf{W}^l), \quad (3)$$

where $\sigma$ denotes the non-linear activation function and $\mathbf{H}^0 = \mathbf{X}$ denotes the initial node attributes. $\tilde{\mathbf{A}} = (\mathbf{D} + \mathbf{I}_n)^{-\frac{1}{2}}(\mathbf{A} + \mathbf{I}_n)(\mathbf{D} + \mathbf{I}_n)^{-\frac{1}{2}}$ represents the normalized adjacency matrix with self-loop.

### 3.3. Transformers

Unlike GNNs that aggregate information from local neighborhoods, Transformers facilitate a holistic aggregation across all token pairs through the self-attention mechanism. The main component of Transformer (Vaswani et al., 2017) is Multi-Heads Self-Attention (MHSA). Given $N$ tokens input $\mathbf{Z} = [\mathbf{z}_i]_{i=0}^{N-1} \in \mathbb{R}^{N \times F}$, which is the concatenation of the initial token embedding $\mathbf{X}$ and the position encoding $\mathbf{P}$ as $\mathbf{Z} = [\mathbf{X}||\mathbf{P}]$, the self-attention first maps the input features $\mathbf{Z}$ to query ($\mathbf{Q}$), key ($\mathbf{K}$), and value ($\mathbf{V}$) vectors. Then, attention scores from query-key pairs are employed to aggregate the value vectors in a weighted manner, operating as a global message-passing mechanism. Specifically, a Self-Attention (SA) module can be formulated as

$$\mathbf{Q} = \mathbf{Z}\mathbf{W}_Q, \quad \mathbf{K} = \mathbf{Z}\mathbf{W}_K, \quad \mathbf{V} = \mathbf{Z}\mathbf{W}_V,$$

$$\hat{\mathbf{Z}} = Attention(\mathbf{Q}, \mathbf{K}, \mathbf{V}) = \text{softmax}\left(\frac{\mathbf{Q}\mathbf{K}^\top}{\sqrt{F}}\right)\mathbf{V}, \quad (4)$$

where $\mathbf{W}_Q$, $\mathbf{W}_K$, and $\mathbf{W}_V \in \mathbb{R}^{F \times F}$ denotes trainable projection matrices and $F$ stands for the feature dimensions. Further details of GNNs and Transformers are provided in the appendix A.

**Graph Transformer.** To employ a Transformer to graph structure data, it is critical to incorporate graph topology appropriately (Zhuo et al., 2025). There are two kinds of strategies (Ying et al., 2021; Rampásek et al., 2022). Some methods directly employ adjacency matrix $\mathbf{A}$ to regularize the Transformer, e.g., element-wise product $\mathbf{A} \odot \mathbf{Sim}^{QK}$. The other methods encode topology structure into position encoding $\mathbf{P}$, e.g., the eigenvector of its Laplacian matrix. Thus, it is evident that the utilization of the Transformer

for graph-data processing calls for a careful handling of graph topology and node representation, a consideration that stands out as especially striking in the context of brain network analysis.

## 4. Issues with Message Passing

This section begins with the method of brain network construction, followed by the analysis of the GNNs and Transformers on the constructed brain network. Finally, experiments are conducted to verify the above analysis.

### 4.1. Brain Network Construction

Refer to (Cui et al., 2022) for the detailed construction process of the brain network from raw data. Here, the final two steps, which are closely related to the brain network $G$, are considered. Firstly, the Brain Region Parcellation segments each subject into ROIs. Secondly, the weights of edges between ROIs are calculated using pairwise Pearson correlation coefficient as

$$r_{xy} = \frac{\sum_{i=1}^{n}(x_i - \bar{x})(y_i - \bar{y})}{\sqrt{\sum_{i=1}^{n}(x_i - \bar{x})^2}\sqrt{\sum_{i=1}^{n}(y_i - \bar{y})^2}}, \quad (5)$$

where $\{x_i, ..., x_n\}$ and $\{y_i, ..., y_n\}$ are two sequences of response values of two ROIs with the same length $n$, and $\bar{x}$ and $\bar{y}$ are the means of sequences. To obtain a robust graph, a threshold is often used to sparsify the edge weights as

$$a_{xy} = \begin{cases} r_{xy}, & \text{if } r_{xy} > \text{threshold}, \\ 0, & \text{otherwise}. \end{cases} \quad (6)$$

The constructed symmetric $\mathbf{A} = [a_{xy}] \in \mathbb{R}^{N \times N}$ is seen as the adjacency matrix of the brain network. Note that the sequence $\{x_i, ..., x_n\}$ is infrequently characterized as the attribute of ROI, since its content is the response values at some moments instead of the essential characteristics of the ROI. Section 4.2 elaborates on how GNNs and Transformers construct node attributes.

### 4.2. Model Analysis

This section analyzes how GNNs and Transformers process brain networks constructed in Section 4.1.

**GNNs for Brain Network.** As shown in Section 3.2, GNNs learn representation by combining graph topology and node attributes, which are two different types of information. Unfortunately, the constructed brain network often lacks essential node attributes. To alleviate this issue, existing GNN-based methods often constructed node attributes from the brain network topology, such as (1) identity matrix, (2) Eigenvectors of the adjacency matrix, (3) node degree, (4) local statistic of degree, (5) adjacency matrix itself or from another correlation measurement (Fig. 1). This strategy

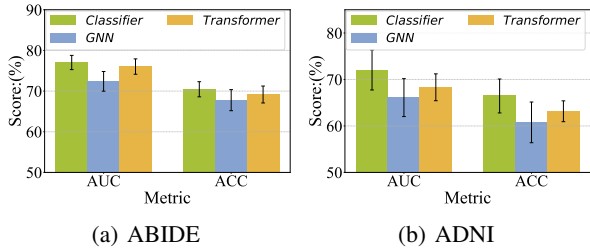

*Figure 2.* Performances of basic classifier, GNN and Transformer on ABIDE (left) and ADNI (right) datasets according to AUC and ACC. Note that the simple classifier consistently outperforms basic GNN and Transformer with message-passing strategy.

goes against GNNs' need for different types of information. Therefore, GNNs' characteristics, especially message passing, can't be fully exploited in brain network modeling.

**Transformers for Brain Network.** As shown in Section 3.3, Transformers learn holistic relations between all tokens with the self-attention mechanism. To employ Transformers in the brain network modeling, the initial token embedding is often set as the adjacency matrix, i.e., $\mathbf{X} = \mathbf{A}$. Since $\mathbf{A}$ is obtained by calculating Pearson correlation coefficients between all ROI pairs, the holistic relationship between all tokens is present in the initial token embedding $\mathbf{X}$. Thus, the necessity of employing Transformers, especially the implicit message passing in the attention mechanism, to explore holistic relations in brain networks is weak (Fig. 1).

In summary, the message-passing under both GNNs and Transformers can't be fully explored and exploited by considering the characteristics of the brain networks.

### 4.3. Experimental Investigation

To verify the above analysis of the ability of GNNs and Transformers in brain networks, this section conducts basic experiments. The performances are measured according to AUC and ACC metrics on ABDIE and ADNI datasets, results of more metrics are placed in Appendix B. Three baselines are involved as follows:

- The simple classifier $f(\mathbf{A}) = \mathbf{AW}$, with $\mathbf{A}$ as the feature matrix and $\mathbf{W}$ as learnable matrix;

- The GNN as in Section 3.2 with $\mathbf{A}$ as topology and node attribute;

- The Transformer as in Section 3.3 with $\mathbf{A}$ as node attribute.

The performances are shown in Fig. 2. It can be observed that the basic and simple classifier consistently outperforms the other two models, i.e., GNN and Transformer. This result meets our analysis in Section 4.2 that message passing under both GNNs and Transformers can't be fully explored and exploited in brain network modeling. Note that some GNN-based and Transformer-based models, such as Brain

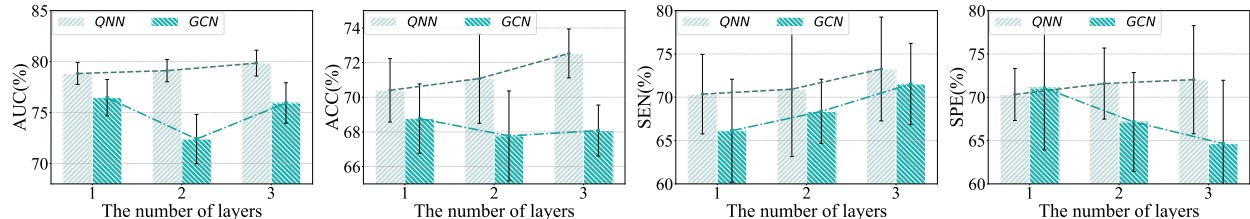

*Figure 3.* Performance comparison of QNN (Hadamard product) and GCN (matrix product) on ABIDE dataset according to AUC, ACC, SEN, and SPE. Note that QNN consistently outperforms GCN, both the AUC and the ACC increase as the number of layers increases.

Network Transformer (Kan et al., 2022b), achieve much better performance compared to the basic GNN and Transformer. However, their high performances mainly rely on the modification to GNN and Transformer instead of the message-passing mechanism behind them.

## 5. Methodology

Brain Quadratic Network (BQN), which breaks the message-passing scheme, is proposed here.

### 5.1. Motivations

The model analysis and experimental investigation in Sections 4.2 and 4.3 demonstrate that message passing under both GNNs and Transformers can't be fully explored and exploited in brain network modeling. Thus, to enhance the performance of brain network analysis, the fundamental brain network operator should be updated.

Here, the simple classifier $f(\mathbf{A}) = \mathbf{AW}$ is employed as the base of the investigation for its satisfactory performance. For one-dimension output, $f(\cdot)$ acts as the linear function as $f(\mathbf{a}) = \sum_{i=1}^{N} w_i a_i = \mathbf{wa}$, where $\mathbf{a} = [a_1, ..., a_N]$ and $\mathbf{w} = [w_1, ..., w_N]$ denote one-row of $\mathbf{A}$ and one-column of $\mathbf{W}$, respectively. It is well-known that linear functions can't implement XOR logic operation (Bishop & Nasrabadi, 2006). Recent attempts (Fan et al., 2018; 2020) demonstrate the expressive ability of the Quadratic/Second-order function of the following forms

$$h(\mathbf{a}) = \left(\sum_{i=1}^{N} w_{ri} a_i + b_r\right)\left(\sum_{i=1}^{N} w_{gi} a_i + b_g\right) + \sum_{i=1}^{N} w_{oi} a_i^2 + c$$
$$= (\mathbf{w}_r \mathbf{a} + b_r)(\mathbf{w}_g \mathbf{a} + b_g) + \mathbf{w}_o(\mathbf{a}^2) + c, \quad (7)$$

where $\mathbf{w}_r = [w_{r1}, ..., w_{rN}]$, $\mathbf{w}_g = [w_{g1}, ..., w_{gN}]$, $\mathbf{w}_o = [w_{o1}, ..., w_{oN}]$, $b_r$, $b_g$, and $c$ are learnable parameters. $\mathbf{a}^2 = \mathbf{a} \odot \mathbf{a}$ denotes the element-wise square operator, and $\odot$ stands for Hadamard product product. It is proved that quadratic function can implement XOR logic operation and possesses some better universal approximation properties compared to linear function (Fan et al., 2020; 2025). This motivates us to employ the quadratic function to enhance the performance of brain network analysis. Moreover, we

provide further elaboration of Quadratic function and its representative applications in Appendix C.

To verify the effectiveness of the quadratic function with the Hadamard product, the following two brain network encoding methods are compared with $\mathbf{H}_0 = \mathbf{AW}^0$ as the initial embedding ($0-$th layer).

- Graph Convolution Network: $\mathbf{H}^l = \sigma(\mathbf{AH}^{l-1}\mathbf{W}^l)$,
- Quadratic Neural Network: $\mathbf{H}^l = \mathbf{H}^{l-1} \odot (\mathbf{AW}^l)$,

where $\odot$ stands for Hadamard product/element-wise product. The performances of the Graph Convolution Network (GCN) and the Quadratic Neural Network (QNN) are shown in Fig. 3. It can be observed that the QNN significantly and consistently outperforms the GCN. Besides, both the AUC and the ACC increase as the number of layers increases.

In summary, both the expressive ability and universal approximation from theory and experimental results suggest the Quadratic Neural Network (QNN) as an alternative for brain network modeling.

### 5.2. Brain Quadratic Network

According to Eq. (7), the Quadratic Network for Brain is formulated by using the Hadamard product as follows.

$$\mathbf{H}^l = \left(\mathbf{H}^{l-1}\mathbf{W}_A^l\right) \odot \left(\mathbf{H}^{l-1}\mathbf{W}_B^l\right) + \left(\mathbf{H}^{l-1} \odot \mathbf{H}^{l-1}\right)\mathbf{W}_C^l, \quad (8)$$

where $\mathbf{W}_A^l$ and $\mathbf{W}_B^l$ are the learnable parameters for the $l$-th layer. To stabilize the learning process, the first term of Eq. (8) is replaced with $\mathbf{H}^{l-1} \odot (\mathbf{AW}_A^l)$, which meets the QNN formula in Section 5.1, and the Brain Quadratic Network(BQN) is as follows:

$$\mathbf{H}^l = \mathbf{H}^{l-1} \odot \left(\mathbf{AW}_A^l\right) + \left(\mathbf{H}^{l-1} \odot \mathbf{H}^{l-1}\right)\mathbf{W}_H^l, \quad (9)$$

where learnable mappings $\mathbf{W}_A^l$ and $\mathbf{W}_H^l$ are for original brain network $\mathbf{A}$ and the embedding $\mathbf{H}$. As shown in the experiments in Section 5.1, the first term in Eq. (9) plays a key role in performance enhancement, while the second one mainly tends to reduce the variance.

### 5.3. Theoretical Analysis

This section provides deep insight into the Brain Quadratic Network, especially the first term in Eq. (9), i.e.,

$$\mathbf{H}^l = \mathbf{H}^{l-1} \odot \left(\mathbf{A}\mathbf{W}_A^l\right). \tag{10}$$

The following theorem provides an intuitive understanding from the concept of community detection.

**Theorem 5.1.** *The iterative formula Eq. (10) of the proposed BQN is equivalent to the updating rule of the following community detection objective function, which is based on nonnegative matrix factorization (NMF) of the adjacency matrix.*

$$\min_{\mathbf{H} \geq 0, \mathbf{W}_A \geq 0} \mathcal{L}(\mathbf{H}) = \|\mathbf{H}\mathbf{W}_A^\top - \mathbf{A}\|_F^2, \tag{11}$$

*where $\mathbf{H} \geq 0$ denotes all elements in $\mathbf{H}$ are non-negative.*

*Proof.* The constrained optimization in Eq. (11) can be converted to an unconstrained one by employing a Lagrange multiplier (Bertsekas, 2014). To this end, a Lagrange multiplier matrix $\mathbf{\Theta}$ is introduced corresponding to the nonnegative constraints $\mathbf{H} \geq 0$. Thus, the equivalent objective function is

$$\min_{\mathbf{H}} \mathcal{L}(\mathbf{H}) = \left\|\mathbf{H}\mathbf{W}_A^\top - \mathbf{A}\right\|_F^2 + \mathrm{tr}(\mathbf{\Theta}\mathbf{H}^\top).$$

Set derivative of $\mathcal{L}(\mathbf{H})$ with respect to $\mathbf{H}$ to 0, it obtains

$$\mathbf{\Theta} = 2\mathbf{A}\mathbf{W}_A - 2\mathbf{H}\mathbf{W}_A^\top\mathbf{W}_A. \tag{12}$$

Following the KKT condition for the non-negativity of $\mathbf{H}$, Eq. (12) can be further reformulated as element-wise form

$$\left(2\mathbf{A}\mathbf{W}_A - 2\mathbf{H}\mathbf{W}_A^\top\mathbf{W}_A\right)_{ij}\mathbf{H}_{ij} = \mathbf{\Theta}_{ij}\mathbf{H}_{ij} = 0, \tag{13}$$

which is the fixed point equation that the solution must satisfy at convergence. Given an initial value of $\mathbf{H}$, the updating rule of $\mathbf{H}$ can be formulated as

$$\mathbf{H}_{ij} \leftarrow \mathbf{H}_{ij} \frac{(\mathbf{A}\mathbf{W}_A)_{ij}}{(\mathbf{H}\mathbf{W}_A^\top\mathbf{W}_A)_{ij}}, \tag{14}$$

where the denominator is to normalize the iterations, and $\leftarrow$ denotes the assignment operation. Therefore, the iteration can be written as in matrix form

$$\mathbf{H} = \mathbf{H} \odot (\mathbf{A}\mathbf{W}_A), \tag{15}$$

which is the same as the iterative formula of BQN. $\qquad\square$

The term $\|\mathbf{H}\mathbf{W}_A^\top - \mathbf{A}\|_F^2$ in Eq. (11) is the nonnegative matrix factorization of the adjacency matrix $\mathbf{A}$. Nonnegative matrix factorization of the adjacency matrix is a widely-used community detection method (Yang & Leskovec, 2013;

Wang et al., 2016; Luo et al., 2022), since its superior performance and outstanding interpretability. Therefore, the formula Eq. (10) leads the proposed BQN to seek representation, which can capture mesoscopic community structure to reflect brain functional modules. This provides an interpretation of the success of the proposed BQN. In addition to the theoretical analysis, experiments reflecting the clustering characteristics of brain region representations obtained through stacking multiple layers of Eq. (9) are presented in Appendix D.

## 6. Evaluations

### 6.1. Experiments Setup

**Datasets.** In the experiments, two real-world fMRI datasets are employed: as follows.

- *Autism Brain Imaging Data Exchange (ABIDE)*: This dataset is primarily utilized to investigate brain functional connectivity variation and structural differences associated with Autism Spectrum Disorder. The pre-processed data version can be accessed from the official website[1]. It collects resting-state functional magnetic resonance imaging data from 17 international sites, as well as anatomical and phenotypic data. The dataset we used contains 516 Autism Spectrum Disorder patients (ASD) and 493 normal controls (NC).

- *Alzheimer's Disease Neuroimaging Initiative (ADNI)*: ADNI is a widely utilized multimodal neuroimaging repository focused on Alzheimer's disease. The raw images can be obtained from ADNI official website[2]. Access is limited and requires adherence to a request procedure to acquire the data. The dataset for this paper contains 53 Alzheimer's disease (AD) samples and 71 normal controls.

The construction of brain functional connectivity matrix for ABIDE is based on Craddock 200 atlas. The Pearson correlation coefficient between the region-averaged BOLD signals from pairs of ROIs (Regions of Interest) is adopted as the measure of functional connectivity strength between ROIs, namely as brain graph adjacency matrix (Cui et al., 2022; Bessadok et al., 2023; Xu et al., 2024). For ADNI, the fMRI data are first preprocessed using the Data Processing Assistant for Resting-State fMRI (DPARSF) toolkit. Next brain ROIs are defined based on AAL 90 atlas and the average time-series feature is calculated for each individual brain ROI. Pearson correlation coefficient between ROIs is then calculated, which serves as the functional connectivity matrix. Specifically, thresholds are set to keep edges with positive weights and drop those with negative weights.

---

[1]http://preprocessed-connectomes-project.org/abide/

[2]https://adni.loni.usc.edu/

*Table 1.* Performance comparison with two categories of baselines. The best model is bolded and the runner-up is underlined, respectively.

| Type | Model | ABIDE | | | | ADNI | | | |
|---|---|---|---|---|---|---|---|---|---|
| | | AUC↑ | ACC↑ | SEN↑ | SPE↑ | AUC↑ | ACC↑ | SEN↑ | SPE↑ |
| Graph Neural Networks | GCN | $59.59_{\pm3.44}$ | $59.30_{\pm3.38}$ | $56.67_{\pm4.37}$ | $61.55_{\pm5.29}$ | $62.45_{\pm3.63}$ | $59.12_{\pm4.53}$ | $54.55_{\pm9.96}$ | $62.00_{\pm8.55}$ |
| | GAT | $60.43_{\pm3.88}$ | $60.10_{\pm4.13}$ | $59.26_{\pm5.51}$ | $62.89_{\pm8.03}$ | $62.00_{\pm2.88}$ | $58.75_{\pm2.86}$ | $53.20_{\pm7.50}$ | $64.79_{\pm7.65}$ |
| | BrainGNN | $64.42_{\pm3.57}$ | $63.09_{\pm1.35}$ | $65.65_{\pm2.88}$ | $60.67_{\pm3.68}$ | $61.81_{\pm1.58}$ | $58.72_{\pm3.14}$ | $52.88_{\pm9.70}$ | $62.70_{\pm4.26}$ |
| | BrainGB | $70.32_{\pm3.66}$ | $65.12_{\pm3.90}$ | $67.01_{\pm10.00}$ | $60.07_{\pm8.53}$ | $66.44_{\pm3.33}$ | $63.70_{\pm4.65}$ | $60.73_{\pm8.25}$ | $64.67_{\pm9.07}$ |
| | FBNETGEN | $74.55_{\pm3.77}$ | $67.09_{\pm3.37}$ | $64.71_{\pm9.85}$ | $69.61_{\pm9.30}$ | $67.05_{\pm2.16}$ | $63.26_{\pm1.38}$ | $66.79_{\pm6.93}$ | $61.31_{\pm9.65}$ |
| | A-GCL | $73.86_{\pm2.91}$ | $\underline{71.04}_{\pm2.40}$ | $71.42_{\pm3.03}$ | $70.95_{\pm3.19}$ | $68.15_{\pm3.33}$ | $67.08_{\pm4.57}$ | $65.45_{\pm8.39}$ | $68.33_{\pm4.36}$ |
| Graph Transformer Models | SAN | $71.35_{\pm2.18}$ | $65.34_{\pm2.91}$ | $55.41_{\pm9.29}$ | $68.39_{\pm7.50}$ | $66.11_{\pm3.41}$ | $61.78_{\pm4.22}$ | $53.94_{\pm7.56}$ | $63.63_{\pm8.51}$ |
| | Graphormer | $63.91_{\pm4.05}$ | $61.88_{\pm6.85}$ | $66.30_{\pm9.98}$ | $55.74_{\pm11.00}$ | $60.69_{\pm5.26}$ | $55.75_{\pm3.18}$ | $60.18_{\pm11.36}$ | $47.75_{\pm13.53}$ |
| | GraphTrans | $60.13_{\pm6.73}$ | $57.83_{\pm4.71}$ | $65.70_{\pm10.30}$ | $49.77_{\pm11.52}$ | $61.41_{\pm3.65}$ | $58.60_{\pm5.41}$ | $65.57_{\pm6.05}$ | $54.37_{\pm3.42}$ |
| | BrainNETTF | $77.93_{\pm1.41}$ | $69.26_{\pm2.26}$ | $65.92_{\pm8.60}$ | $\mathbf{73.20}_{\pm6.06}$ | $69.73_{\pm2.61}$ | $\underline{67.85}_{\pm2.92}$ | $63.64_{\pm6.27}$ | $70.67_{\pm8.33}$ |
| | ContrastPool | $57.36_{\pm0.87}$ | $57.44_{\pm0.69}$ | $57.66_{\pm6.85}$ | $57.08_{\pm7.79}$ | $68.17_{\pm3.28}$ | $66.21_{\pm3.90}$ | $61.51_{\pm7.44}$ | $\mathbf{72.43}_{\pm6.53}$ |
| | ALTER | $\underline{77.99}_{\pm2.21}$ | $70.10_{\pm2.26}$ | $\underline{72.84}_{\pm7.40}$ | $67.68_{\pm5.81}$ | $\underline{71.86}_{\pm3.13}$ | $66.92_{\pm3.93}$ | $\mathbf{71.55}_{\pm8.91}$ | $64.00_{\pm6.80}$ |
| | BioBGT | $69.96_{\pm1.18}$ | $69.70_{\pm2.90}$ | $67.04_{\pm3.41}$ | $72.02_{\pm4.67}$ | $63.16_{\pm3.74}$ | $62.27_{\pm3.23}$ | $63.97_{\pm7.88}$ | $60.55_{\pm6.71}$ |
| | BQN (Ours) | $\mathbf{79.85}_{\pm1.27}$ | $\mathbf{72.53}_{\pm1.41}$ | $\mathbf{73.26}_{\pm5.99}$ | $\underline{72.03}_{\pm6.24}$ | $\mathbf{74.18}_{\pm3.34}$ | $\mathbf{68.62}_{\pm3.22}$ | $\underline{70.91}_{\pm8.60}$ | $68.00_{\pm7.81}$ |

**Baselines.** Thirteen Baselines are compared in the experiments, which can be divided into two categories.

- Graph Neural Network (GNN)-based models, including two typical GNNs: GCN (Kipf & Welling, 2017) and GAT (Veličković et al., 2018)), and four brain-specific GNNs: BrainGNN (Li et al., 2021), BrainGB (Cui et al., 2022), FBNETGEN (Kan et al., 2022a) and A-GCL (Zhang et al., 2023).

- Graph Transformer (GT)-based models, including three typical GTs: SAN (Kreuzer et al., 2021), Graphormer (Ying et al., 2021) and GraphTrans (Wu et al., 2021a)), and four brain-specific GTs: Brain-NETTF (Kan et al., 2022b), ContrastPool (Xu et al., 2024), ALTER (Yu et al., 2024) and BioBGT (Peng et al., 2025).

Additional comparative experiments of BQN are conducted on supplemental datasets, please refer to Appendix E.

**Metrics.** To conduct comprehensive evaluations of performance, a combination of machine learning and medical diagnostic-specific metrics are employed, including Area Under the Receiver Operating Characteristic Curve (AUC), Accuracy (ACC), Sensitivity (SEN), and Specificity (SPE). AUC provides a threshold-independent measure of the model's ability to distinguish between classes. ACC is a straightforward metric to assess the model's classification performance. SEN, also known as the true positive rate, is a critical metric for medical diagnostics. In contrast, SPE represents the true negative rate.

**Implementation Details.** The experiments are performed on a GeForce RTX3090 GPU. For all datasets, we employ random splits with the ratio 7:1:2 to get the training set, validation set and test set. Furthermore, the number of training epochs is set to 200 with batch size 16. An adam optimizer is adopted with initial learning rate of $10^{-4}$ and weight decay as $10^{-4}$ while training, target learning rate is from $10^{-5}$ to $10^{-4}$. The activation function we selected is LeakyReLU. The number of layers, i.e., $k$ is selected from 1, 2, 3, 4, 5 and the dropout rate is chosen from 0., 0.1, 0.2, 0.3. The results were averaged over 5 random runs. Notably, existing studies of GNNs and Transformers for brain networks show marked differences in data processing and model selection, which poses challenges in fairly evaluating their performance. To address this, this paper unifies these settings and compares methods in a fair manner. The details of corresponding unified settings are elaborated in the Appendix F.

### 6.2. Result Analysis

**Brain Disorder Disease Classification.** The performance comparison between BQN and the baseline models on the ABIDE and ADNI datasets are presented in Tab. 1. Upon observation, it is evident that compared to GNN-based and GT-based baselines, BQN exhibits superior performance in terms of AUC and ACC metrics. In particular, on the ABIDE dataset, BQN achieves an accuracy that is 2.43% higher than that of the runner-up baseline ALTER. This advantage can be attributed to the fact that BQN directly leverages the essential information from brain functional connectivity matrices. Therefore, it reduces model complexity and inference uncertainty that may arise from extraneous factors. Moreover, during training, BQN implicitly learns characteristic representations of brain functional modules, as evidenced in Sec. 5.3. These learned representations of functional regions enhance the accuracy of brain disorder classification. By directly modeling the intrinsic properties of brain functional connectivity, BQN is particularly well-suited for brain disorder classification tasks, especially

*Table 2.* Comparison of running time(s) on ABIDE and ADNI.

| Method | ABIDE | ADNI |
|---|---|---|
| GCN | 14.26 | 4.12 |
| FBNETGEN | 19.73 | 5.71 |
| A-GCL | 41.57 | 6.42 |
| SAN | 712.01 | 152.23 |
| Graphormer | 973.52 | 179.11 |
| BrainNETTF | 17.32 | 4.77 |
| ContrastPool | 283.02 | 76.05 |
| ALTER | 37.21 | 7.12 |
| BioBGT | 31.24 | 6.78 |
| BQN(Ours) | **11.31** | **3.33** |

when dealing with the complex and limited-scale nature of neuroimaging datasets. These experimental results and analyses indicate superior performance and highlight the promising potential of BQN.

**Efficiency Test.** This experiment aims to thoroughly examine the efficiency of the proposed BQN. It achieves this by comparing the running time of BQN with those of nine representative baselines. The times reported in Tab. 2 are the total times of training 100 epochs in full batch mode. It can be observed that compared to all baselines, the proposed BQN achieves the shortest runtime on ABIDE and ADNI datasets. This is mainly attributed to the simplicity of the proposed BQN, which guarantees its efficiency. To be specific, the core computation of BQN involves element-wise multiplication of matrices via Hadamard product, which poses a quadratic complexity of $O(n^2)$, where $n$ denotes ROIs number. In contrast, both GNN-based and GT-based models require the computation of matrix multiplication between two square matrices (the propagation matrix and the feature matrix, e.g., Pearson matrices), which typically involves a cubic complexity of $O(n^3)$.

**Additional Experiments w/o Hadamard residual.** To validate the rationality and effectiveness of residuals, i.e., the second term proposed by Eq. (9). An ablation study is conducted with an architectural theme defined as $\mathbf{H}^{(l)} = \mathbf{H}^{(l-1)} \odot \mathbf{A}\mathbf{W}_A^l$ with or without $(\mathbf{H}^{l-1} \odot \mathbf{H}^{l-1})\mathbf{W}_H^l$ moduel, where $\mathbf{H}^0 = \mathbf{A}\mathbf{W}$, $\mathbf{W}$ represents a linear layer. The comparison results are shown in Fig. 4. It can be seen that for the AUC, ACC, and SEN metrics, the addition of the Hadamard residual provides consistent performance improvements on both ABIDE and ADNI datasets. This illustrates the validity of the Hadamard residual. Besides, for AUC and ACC metrics, the standard deviations of the model results are smaller, indicating that the Hadamard residual also contributes to the stability of the model.

**Hyperparameter Sensitivity Analysis.** This experiment is designed to offer an intuitive understanding of how to select the optimal number of layers $k$. The effect of this hyperparameter on the model performance is reported in

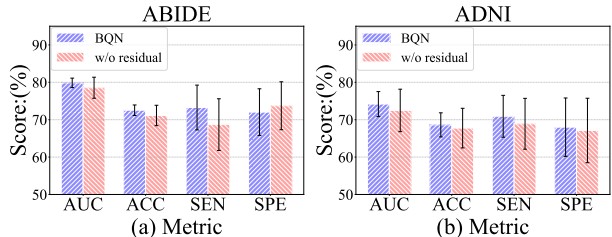

*Figure 4.* Performance comparison of BQN with and without Hadamard residual term on two datasets.

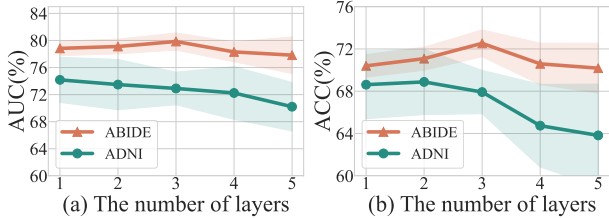

*Figure 5.* Performance variations for varying $k$ on two datasets.

Fig. 5. From this figure, it is evident that the proposed BQN achieves stable performance on these two datasets when the parameter $k$ is within the range $\{1, 2, 3\}$. This demonstrates the stability of BQN. Moreover, the experimental results indicate that the model performance exhibits a gradual decline as the number of layers increases. Considering that brain datasets are generally of limited size (on the order of hundreds of samples), the observed performance degradation can be attributed to overfitting, which is likely induced by the excessive number of parameters and the heightened model complexity associated with deeper architectures.

**Case Study.** To interpret the rationality of BQN on the ABIDE dataset, we construct two contrast brain graphs that reflect brain network differences among distinct groups. The first contrast brain graph is derived from the initial brain connectivity matrices, while another contrast brain graph is from the brain network generated by BQN. Specifically, the brain connectivity matrix template of ASD is obtained by averaging functional connectivity matrices within ASD groups, $A_{\text{Template}}^{\text{ASD}} = \frac{1}{n} \sum_{i=1}^{n} A_i^{\text{ASD}}$. Brain template of NC is obtained similarly, $A_{\text{Template}}^{NC} = \frac{1}{m} \sum_{i=1}^{m} A_i^{NC}$, where $m$ denotes the number of normal subjects, and $n$ represents the number of ASD patients in the ABIDE dataset. The contrast brain graph is then calculated as: $\boldsymbol{A}_{\text{contrast}} = \boldsymbol{A}_{\text{template}}^{NC} \ominus \boldsymbol{A}_{\text{Template}}^{ASD}$, where $\ominus$ performs element-wise subtraction on the two input matrices. The learned contrast brain graph is then obtained by performing the same process as the equations above. The only difference is that the output of the last layer of a well-trained BQN is as brain graphs to construct the learnable contrast graph. Notably, the input of the well-trained BQN is a test set, and the phenotype of the subject is based on the result of the classification head of the model.

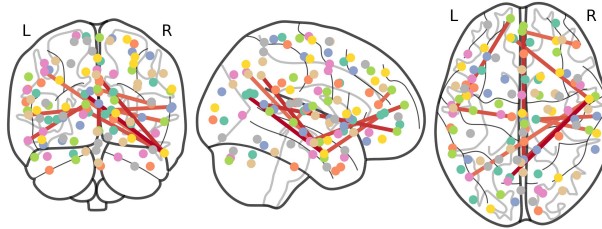

(a) Contrast brain graph constructed from initial data

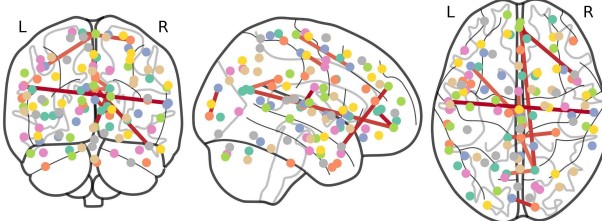

(b) Contrast brain graph constructed from the output of BQN

*Figure 6.* Brain network comparison. Red edges denote significant differences in corresponding brain connectivity between the normal population and individuals with Autism Spectrum Disorder.

The visualization of the two contrast graphs, the top-20 edges with the largest weights are selected, is shown in Fig. 6. Firstly, it can be observed that compared to the initial contrast brain graph (subfigure a), the contrast graph learned by BQN is sparser and retains the core focus of abnormal connectivity, such as connectivity between paracentral lobule region and cingulum region. This finding indicates that the BQN learn a more holistic brain functional connectivity and gain ability of learning characteristic representations of brain functional modules, consistent with the theoretical analysis.

Moreover, our model focuses on the abnormal connections of the prefrontal cortex, cingulate, corpus callosum, and parietal as is shown in Fig. 6. Disruption of functional connectivity in these brain regions has been shown to be strongly associated with ASD patients (Assaf et al., 2010; Weng et al., 2010). This indicates that our model possesses a high degree of biological interpretability, rather than merely an improvement in performance.

## 7. Conclusions

This paper observed that the message-passing mechanism, commonly used in brain network analysis, is redundant from both model analysis and experimental investigation. Based on this finding, BQN is introduced by employing Quadratic Network. BQN is a simple but novel model that adaptively learns brain functional networks while implicitly clustering connectivity between brain regions. Comprehensive experiments on the ABIDE and ADNI datasets demonstrate that

BQN consistently outperforms models based on GNNs and Transformers. We hope this study could offer valuable insights into simpler models that can also achieve remarkable performance in brain disorder disease classification.

## Acknowledgments

This work was supported in part by the National Natural Science Foundation of China (No. U22B2036, 62376088, 62025604, 92370111, 62272340, 62261136549), in part by the Hebei Natural Science Foundation (No. F2024202047), in part by the National Science Fund for Distinguished Young Scholarship (No. 62025602), in part by the Hebei Yanzhao Golden Platform Talent Gathering Programme Core Talent Project (Education Platform) (HJZD202509), in part by the Post-graduate's Innovation Fund Project of Hebei Province (CXZZBS2025036), in part by the Tencent Foundation, and in part by the XPLORER PRIZE.

## Impact Statement

This paper presents work whose goal is to advance the field of Machine learning. There are many potential societal consequences of our work, none of which we feel must be specifically highlighted here.

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

In the appendix, the presented contents are as follows:

- A: Further elaboration on GNNs and Transformers.

- B: More metrics' analysis.

- C: Theoretical and application progress of quadratic networks.

- D: Experiments on the clustering characteristics of node representations.

- E: Further comparative experiments of BQN.

- F: Uniform configurations.

## A. Further Elaboration on GNNs and Transformers

To better illustrate the message passing mechanism of GNNs, we adopt GCN, SGC (Wu et al., 2019), and APPNP (Klicpera et al., 2019) as examples. These models employ the weighted average function to implement Eq. (1) and Eq. (2) as:

$$
\begin{aligned}
GCN: \quad & \mathbf{H}^l = \sigma(\tilde{\mathbf{A}}\mathbf{H}^{l-1}\mathbf{W}^l), \\
SGC: \quad & \mathbf{H}^l = \tilde{\mathbf{A}}\mathbf{H}^{l-1} = \tilde{\mathbf{A}}^l\mathbf{X}, \\
APPNP: \quad & \mathbf{H}^l = (1-\alpha)\tilde{\mathbf{A}}\mathbf{H}^{l-1} + \alpha\mathbf{X},
\end{aligned}
\tag{16}
$$

where $\sigma$ denotes the non-linear activation function (e.g., $\mathrm{ReLU}(\cdot)$), $\alpha$ is the balancing hyper-parameter and $\mathbf{H}^0 = \mathbf{X}$ denotes the initial node attributes. $\tilde{\mathbf{A}} = (\mathbf{D} + \mathbf{I}_n)^{-\frac{1}{2}}(\mathbf{A} + \mathbf{I}_n)(\mathbf{D} + \mathbf{I}_n)^{-\frac{1}{2}}$ represents the normalized adjacency matrix with self-loop. Above three models achieve enhanced neighborhood aggregation and residual combination by employing a meticulously designed message passing function, which significantly improves the model's performance and accuracy in capturing complex patterns and relationships within the data. Recently, GNNs struggle to capture long-range dependencies since stacking multiple layers tends to cause over-smoothing and over-squashing issue (Li et al., 2018; Topping et al., 2022) and leads to the loss of discriminative information.

For the representation learning function Eq. (4) in Transformer Section 3.3, it can be reformulated in token-wise form as

$$
\hat{\mathbf{z}}_v = \sum_{u\in\mathcal{V}} \mathbf{S}^{QK}_{v,u} \cdot \mathbf{v}_u = \sum_{u\in\mathcal{V}} \frac{\exp(sim(\mathbf{q}_v, \mathbf{k}_u))}{\sum_{u\in\mathcal{V}} \exp(sim(\mathbf{q}_v, \mathbf{k}_u))} \cdot \mathbf{v}_u,
\tag{17}
$$

where $sim(,)$ terms the similarity function, which mainly adopts scaled dot-product attention, *i.e.*, $sim(\mathbf{Q}, \mathbf{K}) = \mathbf{Q}\mathbf{K}^\top/\sqrt{F}$. Thus, the attention score matrix $\mathbf{S}^{QK} \in \mathbb{R}^{N\times N}$ acts as the propagation weights, and the Transformer follows the holistic message-passing scheme.

## B. More Metrics' Analysis

Owing to the restricted number of datasets employed and the absence of consistent trends across multiple metrics, we conducted additional experiments to support the conclusions as presented in Fig. 2, 3 and 5.

To evaluate the performance of the basic classifier, GNN and Transformer (Fig. 2), we further conducted experiments on the ABIDE and ADNI datasets, incorporating Precision and Recall as metrics. The results are presented in Fig. 7. These additional experiments reveal consistent trends across the datasets, supporting the hypothesis that the simple classifier outperforms the basic GNN and Transformer with message-passing strategy.

For Fig. 3 (performance comparison of QNN and GCN) and Fig. 5 (performance variations for varying layers), we adopted the micro-F1 score to provide a more comprehensive evaluation of model performance in the above two experiments. The experimental results are presented in Fig. 8. Subfigure (a) shows that the QNN significantly and consistently outperforms the GCN as the number of layers increases. In subfigure (b), BQN achieves stable performance when the number of layers increases, with the best performance occurring at layer 3 on the ABIDE dataset. In contrast, the results indicate a gradual performance decline as the number of layers increases on the ADNI dataset. These experimental trends align with the findings from the main-body experiments in Fig. 3 and 5.

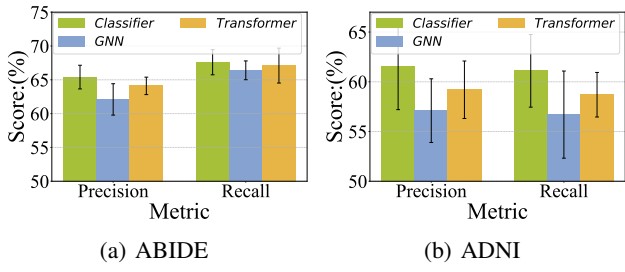

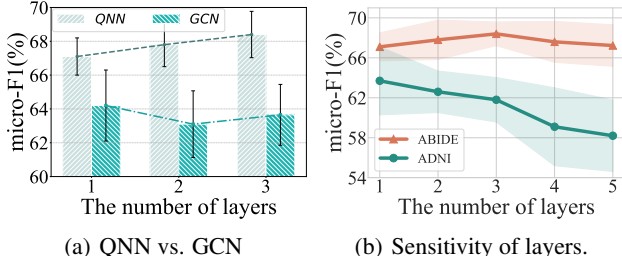

Figure 7. Performances of basic classifier, GNN and Transformer on ABIDE (left) and ADNI (right) datasets according to Precision and Recall. Note that the simple classifier consistently outperforms basic GNN and Transformer with message-passing strategy.

Figure 8. Performance comparison of QNN (Hadamard product) and GCN (matrix product) on ABIDE dataset according to micro-F1 (left), and performance variations for varying k on ABIDE and ADNI datasets according to micro-F1 (right).

## C. Theoretical and Application Progress of Quadratic Networks

The Quadratic Neural Network (QNN) was proposed by authors in (Fan et al., 2018). Subsequent researches have further explored the expressive capabilities of quadratic networks. For instance, (Fan et al., 2020) presented four theorems highlighting the advantages of QNNs over conventional networks in expressive efficiency, unique representation, compact architecture and computational capacity. More recently, (Fan et al., 2025) demonstrated that QNN can implement XOR logic operations and exhibit superior universal approximation properties compared to linear functions.

The findings mentioned above have inspired numerous application-oriented studies. AFT (Zhai et al., 2021) introduces an efficient alternative quadratic function to traditional Transformers that eliminates the need for dot product self-attention, demonstrating competitive performance on all the benchmarks, while providing excellent efficiency at the same time. QCNN (Liao et al., 2023) proposes a novel quadratic convolutional neural network for bearing fault diagnosis, incorporating a derived attention mechanism termed qttention that enhances interpretability and effectiveness. QNN-Bi-LSTM (Keshun et al., 2024) utilize a hybrid model combining a quadratic neural network (QNN) with a bidirectional long short-term memory network (Bi-LSTM) for efficient and interpretable rolling bearing fault diagnosis. Graph Reciprocal Network (GRN) (Yang et al., 2023) treats node attributes and topology of graph as reciprocal elements by regarding nodes as another kind of attribute and employing a novel representation scheme based on Quadratic Networks to achieve fine-grained element-wise product representations.

Previous researches have shown the broad applicability of Quadratic Networks across various domains. However, no studies have yet applied Quadratic Networks-based methods to brain network analysis. Our work fills this gap by showing that Eq. (9) of the quadratic function reveals a clustering effect in brain functional regions. The competitive performance of the Brain Quadratic Network (BQN) presented in Section 6 underscores the potential of Quadratic Networks for brain network analysis and validates its reasonable application in this context.

Table 3. Clustering performance with increasing number of layers

| Metric | init | layer_1 | layer_2 | layer_3 |
|---|---|---|---|---|
| SC↑ | 0.004 | 0.179 | 0.250 | 0.327 |
| CH↑ | 5.746 | 12.357 | 18.877 | 22.528 |
| DB↓ | 4.974 | 3.434 | 2.674 | 1.795 |

## D. Experiments on the Clustering Characteristics of Node Representations

The node representation obtained through iterative learning of the multi-layers of Eq. (9), its clustering characteristic has been proved in Section 5.2 theoretically. In this section, we further experimentally validate its clustering property. Following the segmentation criteria outlined in (Dosenbach et al., 2007) and (Dosenbach et al., 2010), the brain is segmented into corresponding functional regions. To assess the clustering characteristics of BQN, we employ three standard cluster metrics: the Silhouette Coefficient (SC), Calinski-Harabasz Index (CH) and Davies-Binould Index (DB). Results are obtained from both the original data and the outputs of three well trained BQN models, using 1 layer, 2 layers and 3 layers respectively,

which are presented in Tab. 3. It can be observed from the table that the proposed BQN effectively captures the clustering properties of functional brain regions. Notably, the results reveal that clustering performance increases as the number of layers increase when the number of BQN layers is within $\{1, 2, 3\}$, which is consistent with Section 5.3.

## E. Further Comparative Experiments of BQN

Considering the limited number of datasets and related studies used in main body, further comparative experiments are employed on ADHD-200 and PPMI datasets. The details of the two datasets and studies we selected are as follows:

**Datasets.** (1) *Attention Deficit Hyperactivity Disorder (ADHD-200)* dataset is a collaboration of 8 international imaging sites that has aggregated and openly sharing neuroimaging data from children and adolescents diagnosed with ADHD and typically developing controls. The dataset used in this paper includes 459 subjects, comprising 230 typically normal individuals and 229 ADHD patients. (2) *Parkinson's Progression Markers Initiative (PPMI)* dataset aims to identify biological markers of Parkinson's risk, onset and progression. Similar to ABIDE, ADNI and ADHD-200, PPMI is also a multi-site dataset. For this paper, the dataset includes 15 normal controls (NC), 67 prodromal individuals, and 113 Parkinson's disease (PD) patients.

The ROI definition in ADHD-200 dataset is based on Craddock 200 atlas, brain graphs are constructed by computing the Pearson correlation coefficients. Specifically, pearson matrices serve as node attributes and brain functional connectivity matrices. For PPMI dataset, ROI definition is based on AAL 116 atlas and node features, edge weights were preprocessed by authors in (Xu et al., 2023).

**Baseline Methods.** Three supplemental baselines are appended to comparison experiments, including A-GCL (Zhang et al., 2023), AGT (Cho et al., 2024) and BrainMGT (Shehzad et al., 2025). Furthermore, the ALTER employed in the primary text was utilized as a baseline for this comparison experiment, thereby providing stronger contrast. The first two studies are GNNs specially designed for Brain Network Analysis, while BrainMGT and ALTER is based on Transformer.

**Metrics.** AUC, ACC, SEN and SPE are used to evaluate models performance on ADHD-200 dataset, consistent with the ABIDE and ADNI datasets. For PPMI dataset, we utilize Accuracy, Precision, Recall and Specificity to assess the classification performance of all baselines. Besides, model efficiency comparisons, e.g., training time-Time(s), are included on both two datasets.

**Implementation Details.** For the above two datasets, random partitions were used with a ratio of $7:1:2$. The training process was configured for 200 epochs, with the batch size within the range of $\{4, 8, 16\}$. Notably, to ensure a fair assessment of the baselines in conjunction with BQN, the epoch exhibiting the lowest loss on the validation set is selected for performance evaluation on the test set across all models. The Adam optimizer, number of layers, and dropout rate were consistent with the configuration selection used for the ABIDE and ADNI datasets in Section 6.1. Moreover, training times reported in Tab. 4 are the total times of training 100 epochs in full batch mode on each dataset.

**Experimental Results.** The performance comparison between BQN and the baseline models on the ADHD-200 and PPMI datasets are presented in Tab. 4. Note that AGT is specifically developed for multi-class classification tasks, whereas ADHD-200 is a binary classification task. Thus, performance comparison with AGT is limited on PPMI dataset, which has three classes. The results reported in the table indicate that BQN achieved optimal performance in three of the four classification metrics on the ADHD-200 dataset, demonstrating its superiority. On the PPMI dataset, BQN is not superior to AGT but still shows competitive performance. Furthermore, in terms of model efficiency, our approach exhibits the shortest training time across both datasets. Above experimental results highlight the promising potential of our model.

*Table 4.* Performance comparison on ADHD-200 and PPMI. The best and the runner-up models is in bold and underlined, respectively.

| Model | ADHD-200 | | | | | PPMI | | | | |
|---|---|---|---|---|---|---|---|---|---|---|
| | AUC↑ | ACC↑ | SEN↑ | SPE↑ | Time(s)↓ | Accuracy↑ | Precision↑ | Recall↑ | Specificity↑ | Time(s)↓ |
| AGT | - | - | - | - | - | $\mathbf{77.69_{\pm 2.81}}$ | $\mathbf{56.42_{\pm 2.46}}$ | $73.69_{\pm 3.50}$ | $\mathbf{79.37_{\pm 2.27}}$ | $\underline{11.08}$ |
| A-GCL | $74.78_{\pm 4.39}$ | $73.11_{\pm 4.30}$ | $72.04_{\pm 4.68}$ | $73.08_{\pm 4.10}$ | 19.71 | $73.65_{\pm 3.19}$ | $48.99_{\pm 3.71}$ | $70.73_{\pm 3.01}$ | $75.67_{\pm 2.19}$ | 14.12 |
| ALTER | $83.16_{\pm 1.61}$ | $73.48_{\pm 1.34}$ | $74.88_{\pm 6.85}$ | $72.20_{\pm 5.82}$ | $\underline{17.38}$ | $73.91_{\pm 3.13}$ | $50.29_{\pm 3.01}$ | $71.81_{\pm 5.64}$ | $77.36_{\pm 4.22}$ | 11.55 |
| BrainMGT | $78.33_{\pm 1.10}$ | $71.27_{\pm 2.26}$ | $71.09_{\pm 5.63}$ | $\mathbf{73.26_{\pm 3.49}}$ | 26.35 | $70.61_{\pm 2.37}$ | $47.10_{\pm 2.26}$ | $68.45_{\pm 4.81}$ | $74.29_{\pm 6.40}$ | 17.55 |
| BQN (Ours) | $\mathbf{83.34_{\pm 1.13}}$ | $\mathbf{75.68_{\pm 1.95}}$ | $\mathbf{79.73_{\pm 2.27}}$ | $71.63_{\pm 4.87}$ | $\mathbf{6.56}$ | $74.26_{\pm 3.95}$ | $52.30_{\pm 3.75}$ | $\mathbf{73.93_{\pm 4.27}}$ | $76.25_{\pm 3.56}$ | $\mathbf{4.79}$ |

## F. Uniform Configurations

Existing GNNs and Transformers for brain networks are very different in data processing and model selection, which makes it difficult to fairly assess their performance. To alleviate this difficulty, we unify these settings and compare methods in a fair manner. Unifying settings of our work can be concluded as two aspects:

- **Model Selection.** We unify early stopping criteria as the lowest loss on the validation set, which is consistent with BrainGNN and many traditional GNNs and Graph transformers.

- **Data Processing.** We employed a uniform preprocessing approach and brain network construction method for all models on each dataset. Specifically, for the ABIDE dataset, our preprocessing and brain network construction were aligned with those used by BrainNETTF, ALTER and BioBGT. While for the ADNI dataset, we maintained consistency with ALTER's preprocessing and brain network construction methods. For the ADHD-200 dataset, our preprocessing and brain network construction were consistent with BioBGT's methods. Finally for the PPMI dataset, the node attributes and edge weights were preprocessed by authors in (Xu et al., 2023).

