# OpenReview forum: "Do We Really Need Message Passing in Brain Network Modeling?"
_ICML.cc/2025/Conference — ICML 2025 spotlightposter_

### Official Review · Reviewer_ufCW · 2025-03-05

**Overall Recommendation:** 4

**Summary:**

This paper investigates brain network modeling and identifies previous methods' shortcomings, including Graph Neural Network (GNN)-based methods and Graph Transformer (GT)-based methods. Specifically, they often use the Pearson correlation coefficients between pairs of ROIs (Regions of Interest) to construct brain networks, which function as node attributes and graph topology. Based on it, this paper introduces a novel Brain Quadratic Network (BQN) using the Hadamard product instead of the matrix product in previous methods. Extensive comparative experiments demonstrate the effectiveness and efficiency of the proposed BQN.

## update after rebuttal
The authors have adequately addressed my concerns; therefore, I would like to maintain my positive evaluation.

**Claims And Evidence:**

The proposed BQN is well-founded and superior, as evidenced by both theoretical analysis and experimental results.

**Essential References Not Discussed:**

There is no other relevant literature that needs to be discussed.

**Experimental Designs Or Analyses:**

The experiments are validated for their soundness. They are designed using widely recognized datasets and criteria. Performance is verified across diverse datasets, and hyperparameter analyses are carried out.

**Methods And Evaluation Criteria:**

The proposed BQN makes sense for brain network analysis. It is novel in its utilization of a simple yet effective quadratic network.

**Other Comments Or Suggestions:**

1）The statement of matrix multiplication is inconsistent. For example, the notation $\mathbf{A}\cdot \mathbf{X}$ is used in Eq. 3, while $\mathbf{Z}\cdot \mathbf{W}$ appears in Eq. 4.

2）In the caption of Figure 4, QBN is clearly a misspelling.

**Other Strengths And Weaknesses:**

**Strengths**

1）The motivation of this paper is meaningful for brain network analysis. The authors are ambitious and reasonably challenge the rationality of message passing in previous methods.

2）Figure 1 is presented with exceptional clarity and ease of comprehension. Figures 2 and 3 provide clear insights and design motivation.

3）The proposed Brain Quadratic Network (BQN) is simple yet is underpinned by robust theoretical foundations.

**Weaknesses**

1）Reproducibility is a concern. Although the authors claim significant performance improvements over state-of-the-art models with a relatively simple architecture, they have not provided code access, making independent verification challenging.

2）Some of the related work mentioned appears to be extraneous or not directly relevant to the scope of this study. While Graph Neural Networks (GNNs) are foundational models for brain network analysis, it is unclear which specific models, especially SGC and APPNP in Eq. 3, are being utilized in this context.

3) Although the meaning can be understood, the expressions "with" and "without" used in Figure 4 are not conventional. The authors are advised to adopt more standard notations, such as "BQN" and "w/o residual" for clarity.

**Questions For Authors:**

Refer to Weaknesses.

**Relation To Broader Scientific Literature:**

Previous methods have primarily focused on designing specialized GNN and GT models for brain network analysis. This paper challenges this conventional approach, which is innovative and promising. It reveals that existing GNN- and GT-based models essentially rely on the generated Pearson correlation matrices in a redundant manner. To address this issue, this paper proposes an efficient Brain Quadratic Network (BQN).

**Theoretical Claims:**

After conducting a thorough review of the proof, I have essentially confirmed its correctness.

---

> ### Author Rebuttal · Authors · 2025-04-01
>
> > Q1.Reproducibility is a concern. Although the authors claim significant performance improvements over state-of-the-art models with a relatively simple architecture, they have not provided code access, making independent verification challenging.
>
> R1. According to your suggestion, the source code has been made available at [https://anonymous.4open.science/r/BQN-demo](https://anonymous.4open.science/r/BQN-demo) for verification purposes.
>
> > Q2. Some of the related work mentioned appears to be extraneous or not directly relevant to the scope of this study. While Graph Neural Networks (GNNs) are foundational models for brain network analysis, it is unclear which specific models, especially SGC and APPNP in Eq. 3, are being utilized in this context.
>
> R2. We agree that the inclusion of certain related work, such as SGC and APPNP, may not be directly relevant to the scope of this study. While these models are foundational in the context of GNNs and their propagation mechanisms are indeed important for understanding message-passing models, their specific application in brain network analysis is limited. Therefore, we will adjust the manuscript to focus more directly on the models and methods that are most relevant to our study. Specifically, we will move the detailed discussion of SGC and APPNP to the appendix.
>
> > Q3.Although the meaning can be understood, the expressions "with" and "without" used in Figure 4 are not conventional. The authors are advised to adopt more standard notations, such as "BQN" and "w/o residual" for clarity. And in the caption of Figure 4, QBN is clearly a misspelling.
>
> R3. Thanks for your careful review. We will revise the legend in Figure 4 to "BQN" and "w/o residual" as suggested. Additionally, we will conduct a thorough review of the manuscript to correct any misspellings, including the error in the caption of Figure 4.
>
> > Q4. The statement of matrix multiplication is inconsistent. For example, the notation $\mathbf{A} \cdot \mathbf{X}$ is used in Eq.3, while $\mathbf{Z} \cdot \mathbf{W}$ appears in Eq. 4.
>
> R4. Thank you for pointing this out. We will revise the notation to maintain consistency. Specifically, we will update Eq. 4 to use the same matrix multiplication notation as Eq. 3, removing the symbol $\cdot$. This will ensure uniformity in our mathematical expressions.

---

### Official Review · Reviewer_nwPW · 2025-03-12

**Overall Recommendation:** 3

**Summary:**

The paper identifies a limitation in the message-passing framework for brain network analysis and proposes an approach, the Brain Quadratic Network (BQN), to address this issue. BQN demonstrates superior performance compared to standard Graph Neural Networks (GNNs) and graph transformers on widely used brain network datasets, highlighting its potential for advancing graph-based neuroimaging analysis.

Update of review after rebuttal: the authors have addressed my concerns and I have increased my rating.

**Claims And Evidence:**

Appears to be.

**Essential References Not Discussed:**

Many closely related recent works in brain network analysis are not included and compared (see the previous comment on "Experimental Designs Or Analyses").

**Experimental Designs Or Analyses:**

1. More recent and relevant related studies should be included and compared for more convincing conclusions on the superiority of the proposed method. Some examples include
[1] Cho, H., Sim, J., et al. Neurodegenerative brain network classification via adaptive diffusion with temporal regularization. In International Conference on Machine Learning (ICML), 2024.
[2] Zhang, et al. A-GCL: Adversarial graph contrastive learning for fmri analysis to diagnose neurodevelopmental disorders. Medical Image Analysis, 90:102932, 2023.
[3] Ma, Hao, Yongkang Xu, and Lixia Tian. "RS-MAE: Region-State Masked Autoencoder for Neuropsychiatric Disorder Classifications Based on Resting-State fMRI." IEEE Transactions on Neural Networks and Learning Systems (2024).
[4] Shehzad, Ahsan, et al. "Multiscale Graph Transformer for Brain Disorder Diagnosis." IEEE Transactions on Consumer Electronics (2025).

2. It appears that the performance on the two datasets are very different from those reported in the related literature, which raises the doubt of the validity of the results. For instance, the accuracy on ADNI/ABIDE reported on ALTER is around 67%/70% but that reported in the original paper is 74%/77%. Similar issues exist for ContrastPool with even larger gaps. What causes such a large difference in performance?

3. The dataset ADNI used is small with around 100 samples. With such a small sample size, how would the proposed method adequately train a reliable model without severe overfitting? A convergence plot is expected to support this.

4. The authors should provide a detailed description of how they select the readout function for the final classification. Given their claim that BQN operates differently from GNNs and Graphormers, it is unclear whether traditional readout mechanisms remain valid for BQN. A more thorough discussion is needed to clarify whether the chosen readout is theoretically aligned with the proposed framework.

5. For the case study, the authors should clearly explain how they transform the output of the final BQN layer into brain graphs. Specifically, it is unclear whether this conversion is based on a thresholding mechanism, top-k edge selection, or other criterion. Providing such details would enhance the interpretability and reproducibility of the method.

**Methods And Evaluation Criteria:**

The paper uses widely adopted datasets and metrics in brain network analysis for the empirical assessment.

**Other Comments Or Suggestions:**

Nil

**Other Strengths And Weaknesses:**

Strengths:
1. BQN demonstrates strong computational efficiency, making it a promising approach for large-scale brain network analysis.
2. The design of BQN is relatively simple yet effective, striking a balance between model complexity and performance.

Weaknesses:
1. Limited baselines selected.
2. Doubtful inconsistent empirical results.
3. Small dataset used for ADNI. More large-scale datasets should be used.
4. More clarifications on the readout function and the case study.

**Questions For Authors:**

Please see the comments above.

**Relation To Broader Scientific Literature:**

This paper addresses an important open question: Is message passing truly necessary for certain graph-related tasks? Prior research has shown that in heterophilic graphs, message passing in GNNs can sometimes degrade node classification performance. The authors contribute to this discussion by providing both theoretical and experimental evidence in the context of static brain network analysis, suggesting that message passing may not be a crucial component for achieving strong performance in this domain.

**Theoretical Claims:**

Appears to be sound.

---

> ### Author Rebuttal · Authors · 2025-04-01
>
> > Q1. Stacking multiple layers of Eq. 10 is theoretically equivalent to a single-layer formulation. The paper lacks theoretical or empirical justification for stacking multi-layer of Eq. 10 leads to performance gains.
>
> R1. There may be a serious misunderstanding. Stacking multiple layers of Eq. 10 is **NOT** equivalent to a single-layer formulation. Take the case of a two-layer BQN as an example. Formulation $AW_A^1 \odot AW_A^2$ at the vector level is equivalent to $(aw_a^1) (aw_a^2)$, where $W_A^l$ means linear transformation of layer $l$. Subsequently, take the example of $a$ contains two variables, $(aw_a^1) (aw_a^2) = (a_1w_{a1}^1 + a_2w_{a2}^1) (a_1w_{a1}^2 + a_2w_{a2}^2)≠(a_1w_{1} + a_2w_{2})$ is clearly obtained for multiply interaction of variables $a_1$ and $a_2$. Thus, at the matrix level $AW_A^1 \odot AW_A^2≠AW_A$, which means stacking multiple layers of Eq. 10 is not equal to the single-layer formulation.
>
> Besides, the performance gains of multi-layer come from the captured community structure in Eq. 11, which is the objective function of NMF-based community detection. Eq. 14 is the **iterative** updating rule of solving Eq. 11, and is equivalent to Eq. 10. Thus, multi-layers of Eq. 10 is the **iterative** updating rule of solving Eq. 11, which can learn better community structure and guarantee the performance gains.
>
> > Q2. More related studies and large datasets should be included.
>
> R2. We have incorporated the mentioned methods into our effectiveness and efficiency comparison and expanded our evaluation to include two additional datasets (ADHD-200 and PPMI), more large-scale than ADNI. All methods use the same data preprocessing, brain network construction and uniform data partition.
>
> No comparison was made with the work [3] for no open source. AGT requires datasets with more than two classes, limiting its evaluation on the PPMI dataset. The experimental results are shown in https://anonymous.4open.science/r/BQN-demo/figure/Table.jpg
>
> Table reveals that BQN outperforms the baselines on the ABIDE, ADNI, and ADHD-200, demonstrating its superiority. On the PPMI dataset, BQN is not superior to AGT but still shows competitive performance while achieving the best efficiency. This highlights the promising potential of our model.
>
> > Q3. The reported results of ALTER and ContrastPool are different from original papers. ?
>
> R3. Existing GNNs and Transformers for brain networks are very different in data processing and model selection. It makes it difficult to fairly assess their performance. To alleviate this difficulty, **this paper unifies these settings and compares methods in a fair manner**.
>
> - **model selection**. We unify early stopping criteria as the lowest loss on the validation set. This is different from ALTER, which uses the criteria of the highest AUC. This causes the performance gap of ALTER. We also find that the proposed BQN also outperforms ALTER with the criteria of highest AUC.
> - **data processing**. We unify data preprocessing and brain network construction as BioBGT[ICLR'25], ALTER, and BrainNETTF. This is different from the ContrastPool, which follows [5]. This causes the performance gap of ContrastPool. Existing work has demonstrated that different brain network construction methods can lead to different performance.
>
> [5]Data-driven network neuroscience: On data collection and benchmark. NeurIPS, 2023
>
> > Q4. A convergence plot is expected to support a reliable trained model using the ADNI dataset.
>
> R4. The convergence plot, which includes training loss, validation loss, and test accuracy using BQN, is available at https://anonymous.4open.science/r/BQN-demo/figure/BQN_convergence.jpg. The overall trend of the three experimental metrics is steady, despite some fluctuations. This indicate that the model generalizes well without severe overfitting.
>
> > Q5. The authors should clarify the chosen readout function theoretically aligns with BQN framework.
>
> R5. We employ OCREAD, which concatenates the embeddings of cluster centers, as the readout function in our proposed method, consistent with BrainNETTF and ALTER. OCREAD exploits the clustering property for readout, which is consistent with the quadratic network in BQN as shown in Theorem 5.1.
>
> Besides, traditional readout mechanisms, such as MEAN and MAX, remain valid for BQN, since they meet the requirements of readout mechanisms from node embeddings to graph embeddings. However, we believe their performance is not as good as OCREAD, since the clustering property is not considered.
>
> > Q6. Elaborate the transformation from the output of the model to brain graphs and provide thresholding details in the conversion.
>
> R6. Given the output of the final BQN layer $A$, it is first calculated as $A=(A+A^T)/2$. Next, $A_{Template}^{ASD}$, $A_{Template}^{NC}$ and $A_{contrast}$ are obtained by equations in Lines 423, 424 and 428 repectively. Finally, the top-20 edges with the largest weights from $A_{contrast}$ are selected for visualization.

---

> > ### Comment · Reviewer_nwPW · 2025-04-03
> >
> > Thank you for the response and apologies for the misunderstanding on stacking multi-layers of Eq. (10).  I have some further questions:
> > - Q2: Why does BQN perform poorer than AGT on PPMI on most of the metrics? Why are the results of AGT missing on ADHD-200?
> > - Q3: what are the results of BQN and ALTER with the criteria of highest AUC? It is claimed that this paper unifies data preprocessing and brain network construction as BioBGT[ICLR'25] ALTER, and BrainNETTF. Nonetheless, the dataset preparation in these studies does not appear to be performed in a unified way. For instance, ABIDE is parcellated by the Craddock 200 atlas, while ADNI is by the AAL-90 atlas. Different pre-processing tools are adopted for different datasets as well: ABIDE is pre-processed by PCP using five different tools; ADNI is pre-processed by DPARSF. Additionally, the number of samples in ADNI used in this work (124 samples) is significantly smaller than that used in BioBGT (407 samples). Any reason for using a smaller ADNI dataset with 2 classes rather than multiple classes?
> > - Q4: please plot the convergence curve on the training accuracy, validation accuracy, and test accuracy for better assessment of overfitting.

---

> > > ### Author Response · Authors · 2025-04-05
> > >
> > > > Q2: Why does BQN perform poorer than AGT on PPMI on most of the metrics? Why are the results of AGT missing on ADHD-200?
> > >
> > > R2: These two questions share the same reason: **AGT is specifically developed for multi-class classification tasks, while ADHD-200 is a binary classification task**. The superior performance of the AGT on the PPMI dataset can be attributed to its ability to learn temporal relations between the multiple diagnostic groups. In the PPMI dataset, there are three diagnostic groups, and the AGT can learn the temporal relationship between them. However, this ability is not applicable to the ADHD-200 dataset, which consists of only two diagnostic groups.
> > >
> > > Specifically, the AGT has a special group-level temporal regularization module that learns temporal dynamics between diagnostic labels. This ability is implemented by designing a loss function $R_{\text{temp}} = \frac{1}{C-2} \sum_{c=1}^{C-2} \left( d_{c,c+1} + d_{c+1,c+2} - d_{c,c+2} \right)$, where $d_{c,c+1}$ denotes feature distance between group $c$ and group $c+1$. By minimizing the $R_{\text{temp}}$, the AGT learns the temporal dynamic relationship between labels. While Parkinson's disease has a distinct progression state process, thus AGT outperformed the BQN model on the PPMI dataset for the above model characteristic. However, for ADHD-200 dataset(consists of 2 diagnostic categories), minimizing the $R_{\text{temp}}$ implies that there is no difference between the ADHD group and the NC group, which is not reasonable. Therefore, AGT was unable to complete the comparison experiment on the ADHD-200 dataset.
> > >
> > > ---
> > >
> > > > Q3.1: What are the results of BQN and ALTER with the criteria of highest AUC?
> > >
> > > R3.1: We have conducted experiments to compare the proposed BQN and ALTER based on the criterion of the highest AUC. The results at [experiment_results](https://anonymous.4open.science/r/BQN-demo/figure/Table_2.png) indicate that **BQN consistently outperforms ALTER** on the ABIDE, ADNI, ADHD-200, and PPMI datasets.
> > >
> > > ---
> > >
> > > > Q3.2: It is claimed that this paper unifies data preprocessing and brain network construction as BioBGT[ICLR'25], ALTER and BrainNETTF. Nonetheless, the dataset preparation in these studies does not appear to be performed in a unified way.
> > >
> > > R3.2: We employed a uniform preprocessing approach and brain network construction method **for all models on each dataset, rather than across all datasets**. This is to alleviate the issue that different models employ different preprocessing and brain network construction methods on the same dataset, making models comparable on each dataset. Specifically:
> > >
> > > - For the ABIDE dataset, our preprocessing and brain network construction were aligned with those used by BrainNETTF, ALTER and BioBGT.
> > > - For the ADNI dataset, we maintained consistency with ALTER's preprocessing and brain network construction methods.
> > > - For the ADHD-200 dataset, our preprocessing and brain network construction were consistent with BioBGT's methods.
> > >
> > > Thanks for your question which makes the description of the experimental setting more clear. And, we will add them to the paper.
> > >
> > > ---
> > >
> > > > Q3.3: Additionally, the number of samples in ADNI used in this work (124 samples) is significantly smaller than that used in BioBGT (407 samples). Any reason for using a smaller ADNI dataset with 2 classes rather than multiple classes?
> > >
> > > R3.3: The ADNI dataset used in BioBGT is not available on the web and the details of the data selection are not provided. The full ADNI dataset, which is obtained from the authors of BioBGT, contains 538 samples, which are divided into four classes. Since they do not provide the details of the data selection for the classification task on three classes, we employed two classes, i.e., AD and NC, for the experiment in the paper.
> > >
> > > According to your suggestion, we have conducted an experiment on the four-class ADNI dataset, maintaining consistency with BioBGT in processing and brain network construction. The results are reported at [multi-class_experiment](https://anonymous.4open.science/r/BQN-demo/figure/Table_3.png). Among the models evaluated, the proposed BQN achieved the highest accuracy and AUC, indicating its good generalization ability.
> > >
> > > ---
> > >
> > > > Q4: Plot the convergence curve on the training accuracy, validation accuracy, and test accuracy for better assessment of overfitting.
> > >
> > > R4: Following your suggestion, we have revised the convergence plot at [convergence_experiment](https://anonymous.4open.science/r/BQN-demo/figure/BQN_convergence_acc.jpg). The plot provides a comprehensive view of the training accuracy, validation accuracy, and test accuracy over the model iterations.
> > >
> > > Observations from the plot indicate that while the training accuracy exceeds the validation and test accuracies, the latter two do not exhibit a decline after reaching a certain level. This aligns with the case of the loss curves and suggests that the proposed BQN does **NOT** have severe overfitting.

---

### Official Review · Reviewer_fMGi · 2025-03-12

**Overall Recommendation:** 4

**Summary:**

The paper proposes Brain Quadratic Network (BQN), a novel approach for brain network modeling that replaces traditional message-passing mechanisms with quadratic networks and Hadamard products. It shows that BQN outperforms GNNs and Transformers on fMRI datasets, achieving higher accuracy and efficiency. Theoretical analysis reveals that BQN implicitly performs community detection, capturing brain functional modules.

## update after rebuttal

**Claims And Evidence:**

Yes, the claims are well-supported by both theoretical analysis and extensive experiments. Theoretical connections to community detection via non-negative matrix factorization (NMF) validate BQN's ability to capture brain network structures. Empirical results on graph datasets demonstrate superior performance and efficiency compared to GNNs and Transformers.

**Essential References Not Discussed:**

No, the paper does not omit any essential related works.

**Experimental Designs Or Analyses:**

Yes, I checked the soundness of the experimental designs and analyses. The experiments on the ABIDE and ADNI datasets are well-designed, and the results are valid. The comparisons with GNN and Transformer baselines are appropriate, and the performance metrics used are suitable for evaluating the classification tasks.

**Methods And Evaluation Criteria:**

Yes, the proposed Brain Quadratic Network (BQN) and the evaluation criteria using fMRI datasets are appropriate for the problem of brain network modeling.

**Other Comments Or Suggestions:**

Some punctuations are missing after certain formulas, such as Equations 6 and 8.
In Equation 8, it seems that $W_A$is reused.

**Other Strengths And Weaknesses:**

**Strengths**

1) The paper is well-written and easy to follow, effectively conveying its contributions.

2) The paper provides experimental results on benchmark datasets.

**Weaknesses**

1) The parameters $b$ and $c$ in Eq. 7 are not represented or utilized in BQN (Eq. 8). Should the layers of the model use MLP?

2) The results are somewhat insufficient. Given the limited number of datasets used and the lack of consistent trends across multiple metrics, it is recommended that the authors provide additional experimental results for other metrics in Figures 2, 3, and 5.

**Questions For Authors:**

See Weaknesses.

**Relation To Broader Scientific Literature:**

The paper's contributions are well-aligned with the broader literature by extending quadratic networks to brain network modeling, connecting the model to community detection via NMF, and providing a simpler, more efficient alternative to GNNs and Transformers.

**Theoretical Claims:**

Yes, I checked the correctness of the proof for Theorem 5.1, which connects the Brain Quadratic Network (BQN) to nonnegative matrix factorization (NMF) for community detection.

---

> ### Author Rebuttal · Authors · 2025-04-01
>
> > Q1.The parameters *b* and *c* in Eq. 7 are not represented or utilized in BQN(Eq. 8). Should the layers of the model use MLP?
>
> R1. Yes, the matrix $\mathbf{W}$ in Eq. 8 represents a Multi-Layer Perceptron (MLP), and the parameters $b$ and $c$ in Eq. 7 are realized in the model implementation by setting bias=True for the MLPs.
>
> > Q2.The results are somewhat insufficient. Given the limited number of datasets used and the lack of consistent trends across multiple metrics, it is recommended that the authors provide additional experimental results for other metrics in Figures 2, 3, and 5.
>
> R2.According to your suggestion, we have conducted additional experiments to supplement the results in Figures 2, 3, and 5. For Figures 2 and 3, we have incorporated precision, recall and micro-F1 metrics on experiments with the ABIDE and ADNI datasets. The results are available at the following links: [Fig2_ABIDE](https://anonymous.4open.science/r/BQN-demo/figure/Fig2_ABIDE.jpg), [Fig2_ADNI](https://anonymous.4open.science/r/BQN-demo/figure/Fig2_ADNI.jpg), [Fig3_F1](https://anonymous.4open.science/r/BQN-demo/figure/Fig3_F1.jpg). These supplementary experiments demonstrate consistent trends across the datasets. For Figure 5, we utilze the micro-F1 score metric to provide a more comprehensive evaluation of model performance as the number of layers increases. The results can be accessed at [Fig5_F1](https://anonymous.4open.science/r/BQN-demo/figure/Fig5_F1.jpg).
> We believe these additional results enhance the robustness of our findings and address the concerns regarding the sufficiency of the experimental evidence.
>
> > Q3.Some punctuations are missing after certain formulas, such as Equations 6 and 8. In Equation 8, it seems that *$W_A$* is reused
>
> R3. Thank you for pointing this out. We will carefully review the manuscript to ensure proper punctuation is used throughout, especially after equations. We will revise Equations 6 to:
>
> $$
> a\_{xy} =
> \begin{cases}
> r\_{xy}, & \text{if } r\_{xy} > \text{threshold}, \\\\
> 0, & \text{otherwise}.
> \end{cases}
> $$
>
> And for Equation 8, we will revise it to:
>
> $$
> \mathbf{H}^l = (\mathbf{H}^{l-1} \mathbf{W}_A^l) \odot (\mathbf{H}^{l-1} \mathbf{W}_B^l) + (\mathbf{H}^{l-1} \odot \mathbf{H}^{l-1}) \mathbf{W}_C^l,
> $$
>
> For $\mathbf{W}_A^l$ is reused, we use $\mathbf{W}_A^l$, $\mathbf{W}_B^l$ and $\mathbf{W}_C^l$ instead. And we will check variable symbols throughly to correct any reuse errors.

---

### Official Review · Reviewer_TdBX · 2025-03-13

**Overall Recommendation:** 4

**Summary:**

This paper investigates the GNN and Transformer, which follows the message passing pipeline, in brain network modeling. It observes that these two methods can’t enhance the performance compared to the vanilla classifier. Following by the analysis of the weakness of them from the brain network construction, it presents a novel and simple method based on Quadratic Network, i.e., Hadamard product, which has attractive properties, such as efficiency and latent clustering. Experiments verify the statement and the superiority of the proposed methods.

**Claims And Evidence:**

I appreciate that this paper can explore this essential question. It ignores existing methods’ modifications to GNN and Transformer and only shows the abilities of GNN and Transformer.
The claims are supported by clear and convincing evidence:
Both model analysis and experiments demonstrate that message passing is not necessary in brain modeling.
Both theoretical analysis and experiments verify the property of the proposed BQN.

**Essential References Not Discussed:**

Sufficient.

**Experimental Designs Or Analyses:**

The design of experiments on both motivations and evaluation of proposed methods are convincing since the experimental setups are common in this field, including datasets, baselines, and criteria.

**Methods And Evaluation Criteria:**

The model analysis on GNN and transformer provides evidence to question the message passing in brain modeling. The proposed BQN makes sense by theoretical analysis on its clustering property. Its rationality is also guaranteed by the theory progress in the Quadratic Network. I suggest that the authors include them in the manuscript.

**Other Comments Or Suggestions:**

The source code is not provided. I suggest including it in the supplementary material.
Efficiency is a remarkable property of the proposed BQN compared to methods based on GNN and Transformer. However, it is only verified in the experiments. I suggest emphasizing it in the introduction and abstract.
The experimental evidence should be provided to justify its clustering property.

**Other Strengths And Weaknesses:**

None

**Questions For Authors:**

In addition to the points in Other Comments or Suggestions, I have another concern. There are some methods based on prototypes, such as [Kan et al., 2022b], the essence of which is clustering. Why does BQN outperform them?

**Relation To Broader Scientific Literature:**

Most SOTA methods on brain network modeling are based on GNN and Transformer, which follow the message passing mechanism. This paper questions its necessity and presents a simple, efficient, and novel method. I believe it may significantly impact this field. I also think it will motivate us to consider whether GNNs are necessary in many fields.

**Theoretical Claims:**

I checked the theorem and its proof.

---

> ### Author Rebuttal · Authors · 2025-04-01
>
> > Q1. The model analysis on GNN and transformer provides evidence to question the message passing in brain modeling. The proposed BQN makes sense by theoretical analysis on its clustering property. Its rationality is also guaranteed by the theory progress in the Quadratic Network. I suggest that the authors include them in the manuscript and emphasizing model efficiency in the introduction and abstract.
>
> R1. Thanks for your suggestion. We will incorporate the mentioned contents of the Quadratic Network into the manuscript, referencing [1], [2], [3], and [4]. Additionally, we will emphasize the model efficiency in both the introduction and abstract to better highlight the contributions of this paper.
>
> [1]Universal approximation with quadratic deep networks. Neural Networks, 2020
>
> [2]An attention free transformer. arXiv preprint, 2021
>
> [3]Attention-embedded quadratic network (qttention) for effective and interpretable bearing fault diagnosis. IEEE Transactions on Instrumentation and Measurement, 2023
>
> [4]Towards efficient and interpretative rolling bearing fault diagnosis via quadratic neural network With Bi-LSTM. IEEE Internet of Things Journal, 2024.
>
> > Q2. The source code is not provided. I suggest including it in the supplementary material.
>
> R2. According to your suggestion, the source code has been made available at [https://anonymous.4open.science/r/BQN-demo](https://anonymous.4open.science/r/BQN-demo) for verification purposes.
>
> > Q3. The experimental evidence should be provided to justify its clustering property.
>
> R3. In response to your suggestion, we have analyzed the clustering properties of the proposed BQN model. The brain is segmented into functional regions using established criteria from prior research [5][6]. The clustering performance is evaluated using three standard metrics: Silhouette Coefficient (SC), Calinski-Harabasz Index (CH), and Davies-Bouldin Index (DB). Results are obtained for both the original data and the outputs of three well trained BQN models, using 1 layer, 2 layers and 3 layers respectively.
>
> |     | init | layer_1 | layer_2 | layer_3 |
> | --- | --- | --- | --- | --- |
> | SC↑ | 0.004 | 0.179 | 0.250 | 0.327 |
> | CH↑ | 5.746 | 12.3536 | 18.877 | 22.528 |
> | DB↓ | 4.974 | 3.434 | 2.674 | 1.795 |
>
> The results reveal that the proposed BQN effectively captures the clustering properties of functional brain regions.
>
> [5]Distinct brain networks for adaptive and stable task control in humans. Proceedings of the National Academy of Sciences, 2007
>
> [6]Prediction of individual brain maturity using fMRI. Science, 2010
>
> > Q4. There are some methods based on prototypes, such as [Kan et al., 2022b], the essence of which is clustering. Why does BQN outperform them?
>
> R4. The performance superiority of the proposed BQN is primarily due to its quadratic network’s approximation capabilities and robust model architecture.
>
> Firstly, BrainNETTF[Kan et al., 2022b] use the Pearson matrix as the feature matrix, which already captures holistic brain region interactions. This limits the capacity of the Transformer's fully connected graph messaging mechanism. In contrast, BQN learns clustering properties and employs a quadratic network with more general function approximation capabilities, resulting in better ROI and graph representations. Secondly, BrainNETTF is a Transformer-based model with larger parameters, increasing the risk of overfitting on relative small brain datasets. BQN, with fewer parameters, reduces model complexity and inference uncertainty.

---

### Decision · Program_Chairs · 2025-05-01

**Decision:**

Accept (spotlight poster)

**Comment:**

This paper presents a compelling and well-executed study that challenges the dominance of message-passing paradigms in brain network modeling, introducing the Brain Quadratic Network (BQN) as a simple yet powerful alternative. The authors provide both theoretical justification and strong empirical evidence to support their claim that message passing may not be necessary for this task, highlighting the effectiveness of Hadamard-based interactions. The work is thought-provoking, with the potential to shift prevailing assumptions in the field, and the proposed method demonstrates strong performance across benchmarks. Given the clarity of the analysis and the novelty of the insight, this paper is a strong accept and should be considered for a spotlight.